# Forward Osmosis Membranes: The Significant Roles of Selective Layer

**DOI:** 10.3390/membranes12100955

**Published:** 2022-09-29

**Authors:** Miao Tian, Tao Ma, Kunli Goh, Zhiqiang Pei, Jeng Yi Chong, Yi-Ning Wang

**Affiliations:** 1School of Ecology and Environment, Northwestern Polytechnical University, Xi’an 710072, China; 2Singapore Membrane Technology Centre, Nanyang Environment and Water Research Institute, Nanyang Technological University, 1 Cleantech Loop, Singapore 637141, Singapore; 3Beijing Origin Water Membrane Technology Co., Ltd., Beijing 101417, China

**Keywords:** forward osmosis, interfacial polymerization, selective layer, polyamide

## Abstract

Forward osmosis (FO) is a promising separation technology to overcome the challenges of pressure-driven membrane processes. The FO process has demonstrated profound advantages in treating feeds with high salinity and viscosity in applications such as brine treatment and food processing. This review discusses the advancement of FO membranes and the key membrane properties that are important in real applications. The membrane substrates have been the focus of the majority of FO membrane studies to reduce internal concentration polarization. However, the separation layer is critical in selecting the suitable FO membranes as the feed solute rejection and draw solute back diffusion are important considerations in designing large-scale FO processes. In this review, emphasis is placed on developing FO membrane selective layers with a high selectivity. The effects of porous FO substrates in synthesizing high-performance polyamide selective layer and strategies to overcome the substrate constraints are discussed. The role of interlayer in selective layer synthesis and the benefits of nanomaterial incorporation will also be reviewed.

## 1. Introduction

One of the most pervasive problems affecting people throughout the world today is the shortage of clean water. Water purification technologies for wastewater treatment and seawater desalination are essential to solving the water crisis by producing more clean water resources through water recycling, reuse and reclamation. Membrane purification technologies are attractive due to the low cost and versatility in producing high-quality water from various water sources such as surface water, brackish water, seawater, and municipal and industrial wastewater. Pressure-driven membrane technologies such as reverse osmosis (RO) have been widely used for half a century, but they may not be suitable for feedwater of super high salinity and/or high viscosity. As an alternative, forward osmosis (FO) has shown potential in harvesting water from these tough-to-treat feedwaters [1,2].

FO exploits the natural phenomenon of osmosis and utilizes the osmotic pressure gradient across a semi-permeable membrane to drive water transport from a low osmotic pressure feed to a high osmotic pressure draw solution. Despite the advantage of potentially lower electrical energy consumption, FO faces several challenges from the membrane, draw solution and process design aspects. For example, an additional process/post-treatment is required to get the final product water or to recover the draw solution, which largely limited its applications. The draw solution design and selection shall properly match with the process/application, as the FO membrane is not a perfect membrane that rejects all the solutes in the feed/draw solutions. Regarding the FO membranes, it is challenging to fabricate a membrane possessing a highly permeable and highly selective skin layer with a porous, less tortuous, and thin support layer, yet having enough strength to perform in long-term application with regular cleaning.

Significant advances have been made in the past decade in the field of FO [3], and nearly 1650 scientific papers have been published in peer-reviewed journals from 2018 to 2022. Among these, 45 review papers have been published with “forward osmosis” in the title (Figure 1). These review papers cover the topics of membrane design [4,5,6,7,8,9,10,11,12,13,14,15,16], fouling mitigation [17,18,19,20,21,22], draw solutes design and recovery [23,24,25,26,27,28], system design [29,30,31,32,33], and applications [34,35,36,37,38,39,40,41,42,43]. Prior studies on FO membrane fabrication placed a lot of focus on porous substrate optimization to reduce the internal concentration polarization (ICP), a factor that causes a significant loss of driving force. Despite various FO membrane fabrication methods and new materials developed, there seems to be a gap between the membrane design and FO applications. When evaluating the membrane performance, many studies only reported basic parameters such as water flux and back solute diffusion. Very few papers have looked into the applications that FO can advantageously serve and the membranes required for those applications. Until today, the direction and strategy of FO membrane development, by considering the realistic and promising FO applications and processes, is still unclear. Some methods may lead to an increase in membrane water permeability coefficient by sacrificing the solute rejection, but most FO applications demand more on high rejection rather than high water permeability.

To bridge the gap between FO membrane development and applications, this study reviews the promising FO applications and state-of-the-art FO membranes and suggests the directions of the future trends for FO membrane design.

### 1.1. Basic FO Concept

Osmosis occurs when a selectively permeable membrane separates two solutions containing solutes of different concentrations. Driven by the osmotic pressure difference, small sized solvent molecules pass through the membrane from the low-concentration side to the high-concentration side. This FO process will continue until an equilibrium is attained, as illustrated in Figure 2a. The Jacobus van ‘t Hoff equation can be applied to quantitate the osmotic pressure from solute concentration as shown in Equation (1):(1)π=icRT
where *π* is osmotic pressure, *i* is the dimensionless van ‘t Hoff index, *c* is the molar concentration of solute, *R* is the general gas constant, and *T* is the absolute temperature in kelvins.

In a more practical FO application, the process includes a high-efficiency membrane separation unit and a draw solution recovery system, as illustrated in Figure 2b, since the draw solute needs to be regenerated for most of the cases (unless the diluted draw solution is the final product). Although a low or no hydraulic pressure is required for the FO separation unit, the draw solutes regeneration may consume a considerable amount of energy and a careful design and selection of draw solutes is needed for an efficient FO process.

### 1.2. FO Target Applications

Compared with traditional pressure-driven processes, FO has been proposed for treating complex wastewaters, including saline wastewater [45,46], industrial wastewater with high salt concentrations from oil and gas [2], mining and metallurgy [47], and cooling tower blowdown [48], streams that need to be concentrated such as liquid food [34,49], sludge [50], and nuclear wastewaters [51]. In general, they can be summarized into two types of applications: general applications that require draw solutes regeneration and special applications without the need for draw solutes recovery (Figure 3). For general application, considering the energy used for draw solutes regeneration, it is believed that the concentration and purification of valuable products and the treatment of high salt concentration water/wastewater (which cannot be treated by RO) are of more interest. The FO process has demonstrated profound advantages in treating feeds with high viscosity, such as food processing [34,36,52]. Without increasing temperature and pressure, the FO process can preserve precious essence and nutrients of products. The DS for this process can be nontoxic inorganic salts such as NaCl, which will be regenerated using the thermal evaporation method. On the other hand, highly saline water (e.g., >7% total dissolved solids (TDS)) can be treated by FO by adopting a thermo-responsive DS with high osmotic pressure. This FO process requires less energy as compared to traditional thermal methods. Since the thermo-responsive draw solute does not undergo phase change or need a lower temperature for phase change (compared to 100 °C for evaporating water) during its regeneration upon heating, the thermal energy needed is less than the thermal methods such as multi-stage flash distillation (MSF) and multi-effect distillation (MED) [53,54]. For special applications, it is not easy to find a draw solute that does not require regeneration. So far, one successful example is the concentrated fertilizer draw solution that can draw water from wastewater for irrigation [1,2,55,56].

Most applied research on FO stays at laboratory-scale feasibility level, as well as demonstration studies which fail to consider long-term fouling, draw solute leakage and the cost and energy associated with the draw solutes regeneration. The limited number of pilot studies [49] could not provide very useful guidance on the membrane development direction. The gap exists between FO applications and membrane advancement. To close this gap, the membrane researchers and manufacturers shall well understand what the promising FO applications are and design the membranes that are suitable for the applications.

### 1.3. Development of DS

Draw solutions with high osmotic pressure, low back diffusion rate and easy regeneration features are crucial in extending the applications of FO technology. Especially, the DS regeneration directly determines if a FO process is feasible. Based on the regeneration methods, the draw solutes can be categorized into non-responsive and responsive ones (Table 1). The most commonly used natural non-responsive draw solutes are inorganic salts, such as NaCl, MgSO_4_, and MgCl_2_. Due to their simplicity and easy availability, they are utilized to evaluate membranes’ properties such as structural parameters [23]. Owing to the small size and high diffusivity, they can generate high FO water flux, but face the issue of reverse diffusion to the feed stream. The recovery of these small-size draw solutes is challenging, as the regeneration methods including membrane separation (e.g., Nanofiltration (NF), RO or membrane distillation (MD)) and thermal evaporation are energy intensive [28]. Many research studies attempted to find a slightly larger draw solute, to balance the membrane rejection and osmotic pressure/diffusivity, so that the draw solute can be recovered by a low-pressure NF or ultrafiltration (UF) process. It is suggested that a draw solute with a molecular weight of about 1000–3000 Da with a narrow polydispersity index (PDI) is quite ideal. In order not to avoid limiting the regeneration method to membrane separation, responsive draw solutes have been suggested or synthesized to utilize magnetic or thermal method for the separation. For example, thermo-responsive draw solutes can precipitate or change phases upon heating, and the separation from water can be realized spontaneously [57,58]. Since low-grade waste heat can be used, the thermal energy for thermo-responsive draw solute recovery is much less of a concern. However, polymer-based draw solutes usually contain small solutes of molecular weight of a few tens to a few hundred, which can penetrate through the FO membrane easily. Although they are just a small amount for a polymer with a low PDI, their reverse diffusion to the feed side still leads to membrane fouling [57]. Hence, a tight FO membrane is usually preferred no matter whether inorganic solutes or polymer-based solutes are used as draw solutes.

## 2. Selecting the High-Selective FO Membranes

### 2.1. State-of-the-Art FO Membranes and Performance Evaluation Method

Generally, based on the structural difference, there are two types of FO membranes: integrally skinned asymmetric (ISA) and thin-film composite (TFC) membrane, as depicted in Figure 4. The ISA membranes, prepared in one-step with an asymmetric structure, have limitations in tailoring the membrane structure independently and normally have a lower water permeability [73]. Cellulose triacetate (CTA) is the most commonly used material to fabricate the ISA type membrane and commercial products are available from Fluid Technology Solutions and Toyobo. The ISA type is much less studied in the literature compared to the counterparts TFC type. TFC membranes on the other hand are prepared in two steps with a layered structure (Figure 4b). Their selectivity and permeability can be tailored by regulating the substrate structure and the selective layer properties, respectively. Not only are they studied by a lot of researchers in the lab, but they are also commercially available from many companies, such as Aquaporin (Singapore), Aromatec (Singapore), Toray (Japan), etc. [74,75].

We summarized state-of-the-art ISA and TFC FO membranes in Table 2 with their FO performance reported in active-layer-facing-feed-solution (AL-FS) orientation. In addition to the water flux (J_v_) and solute flux (J_s_) tested under FO mode, water permeance (*A*) and solute permeability coefficient (*B*) measured under RO mode are often reported to understand the rejection layer properties. Although FO membranes are generally operated under no/low pressure, it is still recommended to measure the *A* and *B* values with low-pressure (e.g., 1–5 bar) RO test to ensure the membranes’ general handleability. In contrast, the FO test is less sensitive in detecting the breakage of the membrane during the measurement, as the draw solute diffusion is from the reversed direction of water flux. Although a thin and porous substrate (i.e., a smaller structural parameter (*S* value) is preferred for reducing the ICP phenomenon, it still needs to have enough mechanical strength for real FO applications. On the other hand, the FO test is usually conducted without monitoring of the pressure, but sometimes a low-pressure gradient exists at a high crossflow velocity, resulting in an inaccurate measurement of FO water flux, especially for a membrane with a large *A* value. With the knowing *A* and *B* values from RO results, we can better design FO experiments and predict FO performance, based on the *B*/*A* and J_s_/J_v_ relationship [76,77].

The *S* value is usually calculated from the *A*, *B* and J_v_ results [78]. The TFC membrane generally has a smaller *S* value and higher *A* value as compared with that of ISA membranes (Table 2), due to the 2-step preparation of substrate and rejection layer that enables the optimization of the two layers. As another important parameter, *B* value is often paid less attention to, as there seems no standard or guideline for the solute rejection of FO membranes. For the RO membrane, the *A*/*B* ratio is a good indicator of the tightness of the rejection layer. This ratio can also apply to FO membranes. It can be seen from Table 2 that *A*/*B* values from <1 to ~20 can be found for various FO membranes. A large *A*/*B* ratio (e.g., >10 bar^−1^) suggests a relatively tight membrane. A small *A*/*B* ratio (e.g., <3 bar^−1^) corresponds to a relatively loose membrane, and the small solute such as NaCl can pass through the membrane more easily, resulting in a greater loss of draw solute and contamination of feed stream/draw solution.

Many studies have focused on overcoming the selectivity-permeability trade-off. To further enhance the membrane performance, nanomaterials such as carbon nanotubes (CNTs), graphene oxide (GO), metal-organic framework (MOF) [79,80], covalent organic framework (COF) [81] have been incorporated into the selective layer of TFC membrane to form thin film nanocomposite (TFN) membranes [82]. From Table 2, it seems that TFN membranes and biomimetic membranes have increased *A* value, which could result in a slightly larger *A*/*B* ratio if the increase in *B* is less significant than the increase in *A*. Nevertheless, the *A* value becomes unimportant for the condition when the feed contains high TDS [83], which will be further discussed in the next section.

### 2.2. High Selective FO Membranes for Targeted Applications

Figure 5 presents FO membranes/modules suitable for different feed solutions. As mentioned earlier, promising FO applications include treating feeds with high TDS and/or high viscosity. Wastewater such as brine, cooling tower blowdown, oil & gas wastewater, and mineral wastewater (e.g., mining) contain a large amount of TDS, and can hardly be treated by pressure-driven membrane processes due to extremely high osmotic pressure. To achieve zero liquid discharge for these wastewaters, FO can be utilized to concentrate till 150–200 g/L TDS using a thermo-responsive draw solute [57]. However, a feed with high TDS content results in a very low FO water flux due to the concentrative concentration polarization, rendering the *A* value and *S* value less important [83,92]. In other words, membranes used for such applications do not necessarily possess a high *A* value and low *S* value. Nevertheless, a low solute passage (high *A*/*B*) is always preferred for all FO applications. On the other hand, a highly viscous feed solution such as liquid from the food industry may or may not have high TDS content. For the feed with low TDS content, a FO membrane with a high *A* value, high *A*/*B* ratio and low *S* value is still the best choice for lowering the concentration of draw solution.

In terms of the membrane module, spiral wound, plate and frame, and hollow fiber configurations are available for FO membranes. For feed with high TDS and low viscosity, spiral wound and hollow fiber modules can be applied with maximized membrane packing density; while plate and frame and hollow fiber modules work better for viscous liquid, for reducing membrane fouling. For long-term running, membrane/membrane modules should have anti-fouling properties and sufficient mechanical strength [93]. The design of the membrane module shall consider hydrodynamic conditions required the minimize membrane fouling for handling these difficult feed streams. Membrane cleaning is a common practice to ensure the sustainability of the process, and the mechanical stability of FO membranes should not be compromised despite their relatively porous support layer (compared to RO membranes).

## 3. Selective Layer of FO Membranes

### 3.1. Substrates and Polyamide Formation

The highly porous substrate with a low structure parameter could mitigate the ICP, which is the key to the success of FO membranes. However, these porous supports could pose challenges in synthesizing high-performance polyamide selective layer. Substrates are the reaction carrier in the interfacial polymerization to synthesize the polyamide layer of TFC membranes. The physical-chemical characteristics of the substrates such as surface porosity, pore size, hydrophilicity, etc. could directly affect the reaction interface, the enrichment of aqueous monomers, the uniformity of monomer dispersion, and the diffusion of monomer to the interface [94]. These will consequently affect the thickness, morphology and crosslinking of the polyamide thin films obtained, and eventually the performance of the membranes. Substrates with high surface porosity and smaller pore sizes tend to facilitate the formation of dense polyamide separation layers with high rejection properties [94], but these surface properties are difficult to achieve in the highly porous FO membrane substrates.

FO membrane substrates are commonly fabricated via non-solvent induced phase separation (NIPS) or electrospinning. To obtain highly porous NIPS substrates, besides tuning the polymer compositions and NIPS conditions, some studies have used methods such as sacrificial macropore templates and incorporation of nanomaterials [95,96]. The additives added could also further tailor the substrates to be more hydrophilic and with controllable pores [97]. On the other hand, electrospun nanofiber membranes have the advantages such as large surface porosity, high porosity and interconnected pore structure, and benefit enormously from the nanofiber-based architectures [98,99,100]. Incorporation of nanofillers in polymeric nanofibers, such as MWCNTs [101], silica [78], bentonite [88], etc., could further enhance the porosity and mechanical strength and tailor the hydrophilicity, alignment and roughness of the nanofibrous support. However, this strategy typically results in increased epidermal pore size and the formation of a selective layer with moderate low selectivity.

Overall, studies on obtaining highly porous substrates that are also suitable for synthesizing high performance of polyamide layer are lacking as almost all focus has been put on addressing the ICP issues. Most FO substrates in recent studies already have considerably high porosity, further increase in the porosity may not see significant improvement in the permeability, and an S value of <500 μm should be sufficiently low to mitigate ICP in most cases. More attention should be given to improving the substrate surface properties, such as surface pore size and hydrophilicity so that a polyamide layer with high selectivity can be obtained.

### 3.2. Role of Interlayers

As mentioned previously, both NIPS and nanofibers have their limitations in tuning the surface pore size and distribution of the substrates for the formation of high selective polyamide layer. Another imaginary strategy is to construct a highly porous intermediate/interlayer on top of the substrates to facilitate the formation of a defect-free and high-performance polyamide layer [102,103,104,105]. There are two main types of intermediate layers: nanomaterial layers and non-nanomaterial layers, according to the materials and construction methods. Nanomaterial intermediate layer made of MOF [79], GO-based materials [15,106,107,108], MoS_2_ [109], oxidized-CNTs mixed layer [110,111], titanium dioxide (TiO_2_) and nanotubes [104,112] have been successfully constructed in previous studies using methods such as vacuum filtration. Though the initial focus was mainly to increase the water flux by providing additional channels for water molecule transport, this intermediate layer will also serve as a surface modifier to provide a favorable surface for polyamide synthesis. The excellent surface hydrophilicity, high porosity and smaller pore size could effectively control the adsorption/diffusion of amine monomers during interfacial polymerization and result in thinner and denser polyamide layers [113,114,115]. Zhang et al. designed a TiO_2_/CNTs nanocomposite intermediate layer on a porous ceramic substrate to help form a defect-free nanovoid-containing polyamide layer with high crosslinking. Compared with the control membrane without an interlayer, the water permeability and NaCl rejection of the resulting FO membrane increased simultaneously, from 1.3 to 2 Lm^−2^ h^−1^ bar^−1^, and from 92.2% to 98%, respectively [104]. On the other hand, non-nanomaterial interlayers are generally composed via coating and cross-linking. For example, the tannic acid-Fe^3+^(TA-Fe^3+^) interlayer was synthesized by coordinating tannic acid using ferric (Fe^3+^) ions as cross-linker on a highly porous substrate [116]. Compared to vacuum filtration for nanomaterial intermediate layer construction, the non-nanomaterial intermediate layer is more versatile, facile, and easy to scale-up, giving controllable chemical and physical structures to TFC FO membranes.

The membranes fabricated with the nanochannel interlayer have outperformed the FO membranes reported so far, providing a new strategy for fabricating high-performance FO membranes using seawater desalination. As an additional step is required during the membrane fabrication, the method used to synthesize the interlayers should be simple and scalable. Spray coating [103,117], brush-painting [105,118], inkjet printing [119] and electrospinning [120] are some encouraging techniques presented in recent studies that have the potential to translate this technology into reality [121,122]. These typical strategies are illustrated in Figure 6.

### 3.3. Suitable IP Methods and Formula

In addition to the porous substrate structure and the intermediate layer, the characteristics of monomers and reaction mediums are critical to the formation of high selective polyamide layer. A typical interfacial polymerization reaction is shown in Figure 7, where m-phenylenediamine (MPD) in the water phase and trimesoyl chloride (TMC) in the oil phase react at the oil-water interface. MPD is a commonly used amine monomer for synthesizing polyamide films with a solute rejection. Many excellent works have focused on improving the separation properties of the polyamide selective layers by adjusting the interfacial polymerization by adding co-solvent [123], zwitterions [124], surfactants, etc. [94]. Many of these studies have synthesised high-rejection polyamide for RO application but similar strategies can also be applied in developing FO membranes with a high *A*/*B* value. This article will only briefly discuss some common strategies and readers can refer to the literature for more details [125,126,127].

The selectivity of the polyamide selective layer is largely determined by the inherent crosslinking density and free volume. The proportion of crosslinked structures can be promoted by adding additives such as dimethyl sulfoxide (DMSO), formamide, acetamide, cyclohexanone, anisole and benzonitrile and 1-methylimidazole during interface polymerization [128,129]. For example, the addition of 1-methylimidazole in the aqueous phase can react with TMC to reduce the thickness of the polyamide layer, make it denser, and enrich the carboxylic acid groups on the surface, achieving a water flux of 72 Lm^−2^∙h^−1^ and rejection of 99.06% using 2000 mg/L NaCl as feed and at 15.5 bar pressure [130]. Another method to improve the retention rate of the membranes is by post-treatments such as heat treatment, secondary crosslinking, coating, etc., but these methods are likely to sacrifice the water permeability [131,132,133]. It should be noted that the NaCl rejection rate of the SWRO membrane can reach 99.5%, which is much higher than that of almost all FO membranes. The methods and formula of interfacial polymerization used in RO membranes can serve as useful clues in obtaining FO membranes with high selectivity. While choosing the suitable formula for interfacial polymerization, it is perhaps also worth thinking about how the formulation can help compensate for the limitation of the porous substrates and achieve a denser polyamide with a high rejection.

### 3.4. Roles of Nanochannels of TFN Membranes

In recent years, nanomaterials with excellent selectivity or water transport channels were compounded with a polyamide layer to prepare thin film nanocomposite (TFN) membranes with enhanced perm-selectivity. These nanomaterials include GO [134], CNTs [135], halloysite/graphitic nitride nanoparticles [136], MOF [137], COF, polyoxometalate based open frameworks (POM-OFs) [138], hydrophilic functionalized titanate nanotubes [139], mesoporous silica [140], aquaporin [141,142], etc. For example, the incorporation of nanomaterials in TFN FO membranes has been demonstrated to effectively increase the permeability of polyamide layers, though often at the expense of selectivity [143]. By adding 2D-MOF nanosheets in the polyamide layer, the water permeability of the TFN membrane was successfully increased by 2.5 times, from 2.1 to 5.0 Lm^−2^ h^−1^·bar^−1^, and the NaCl rejection rate only decreased slightly from 99.3% to 99.2% [144]. By comparing the A vs. A/B value of TFC and TFN membranes using datasets from the reverse osmosis membrane database (https://openmembranedatabase.org/reverse-osmosis-database, accessed on 5 September 2022) [145], as shown in Figure 8, the performance of TFN membranes is unfortunately inferior to those of TFC membranes. The observation deviates from previous expectations as the performance was not improved much statistically. The lower selectivity of TFN membranes could be due to: (1) interfaces/interphases gaps created between nanomaterials and the polyamide; (2) the uneven distribution of the nanomaterials; (3) the defects caused by the aggregation of nanomaterials. To improve the rejection properties, the compatibility of the nanomaterials and the polymer matrix, polyamide layer, and the techniques to effectively disperse the nanomaterials have to be improved so that the sieving properties of these nanomaterials can be fully harnessed [81,146,147,148,149]. For example, COF nanofiller were grafted with hydrophilic carboxyl groups to eliminate particle aggregation and interfacial microvoids between nanofiller and polymeric matrix. With 0.1 mg mL^−1^ of the modified COF, the designed TFN membrane exhibited an 88% improvement in water permeability (from 1.3 to 2 Lm^−2^ h^−1^ bar^−1^), and a 14.3% improvement in selectivity (from 10.77 to 12.3 bar^−1^) compared to the pristine TFC membrane [81].

To have a better distribution of the nanomaterials, vacuumed filtration and spray coating have been previously used [150]. As illustrated in Figure 9a, COFs could be deposited on the substrate by vacuum filtration before interfacial polymerization [151]. The polyamide incorporated with uniformly aligned COF exhibited an enhanced water flux of more than 23% compared to the pristine membrane, without sacrificing its selectivity. Similarly, GO nanosheets with uniform and small size incorporated TFN membrane prepared in the same way, the polyamide layer incorporated with horizontally aligned GO nanosheets exhibited a high water flux at 39.0 L m^−2^ h^−^^1^ and a low specific reverse solute flux at 0.16 g L^−1^, using a 1 M NaCl draw solution [107]. However, the vacuum-assistant method may be difficult to translate to large-scale membrane fabrication. Spray coating can be a promising solution for effectively distributing the nanomaterials and reducing the aggregates at a large scale, as the example shows in Figure 9b [117]. Spraying can ensure precise control of liposome volume and make full usage of liposomes by reducing 2 orders of magnitude without any waste compared to traditional embedding methods. The best membrane incorporated with 4 mg/m^2^ liposomes exhibited a permeability of 3.24 Lm^−2^ h^−1^ bar^−1^ and a NaCl rejection of 99.3%, which is a 27% increase in water permeability compared to the liposome-free membrane (2.56 Lm^−2^ h^−1^ bar^−1^). For other expensive nanomaterials with selective channels, spray can maximize the advantages of these materials, realize industrial production, and greatly reduce material waste during processing. Another concern of TFN membranes is the nanoparticle leaching issue under long-term usage. More consideration should be given to the long-term safety of nanomaterials and new environmentally friendly nanomaterials with good biocompatibility.

### 3.5. Non-Polyamide Selective Layer

The polyamide-based TFC/TFN FO membranes have demonstrated promising performance thus far, though it is difficult to break the selective-permeability trade-off of the material. Additionally, polymer-supported polyamide FO membranes may not withstand challenging feeds that contain strong organic solvents and strong oxidizing agents, or with extreme pH conditions [152,153,154]. A non-polyamide selective layer fabricated by nanomaterials, such as MOFs, 2D carbon-based materials [48], and COFs have been developed for the FO process. For example, a free-standing UiO-66 membrane with a thickness down to 400 nm was successfully fabricated to reduce the S value to 6 µm (Figure 10a) [155]. Compared to TFN membranes, the functionalities of the nanomaterials can be fully harnessed in the continuous MOF crystal layer. However, the UiO-66 membranes only had a water permeability of 1.4 Lm^–2^ h^–1^ bar^–1^ due to the dense layer and an NF-like rejection, Na_2_SO_4_ rejection of 83%. Alongside that, ZIF-8 membranes supported on hollow alumina fiber were recently fabricated via electroless deposition (ELD) of ZnO, followed by a solvothermal synthesis (Figure 10b) [156]. The membrane showed a high water flux of 12.3 Lm^–2^ h^–1^ but a high reverse solute salt flux of 29 gm^–2^ h^–1^ when 10% NaCl solution was used as the draw solution. These new materials could be more chemically stable under harsh conditions compared to polyamide, but their selectivity has to be further improved for practical FO applications as RO-like rejection is highly desirable [157,158,159,160].

## 4. Conclusions

Over the past few decades, FO has aroused great interest in both academia and industry. This review provides the fundamentals of FO, summarizes the state-of-the-art FO membranes and promising practical applications, highlights the important properties an FO membrane shall have and discusses the strategies for developing high-rejection FO membranes. We make a summary and propose future research interests related to FO technology.

(1)In addition to special FO application that does not need DS regeneration, the general FO applications requiring draw solute recovery is promising for feed streams with high TDS and/or high viscosity. For the FO application process that needs to recover the draw solutes, the key to the success of FO is whether the draw solutes can be recovered with low energy consumption and how to avoid or reduce the influence of reverse solutes permeation on the feed solutions;(2)Since the feed stream contains high TDS for most of the promising FO applications, the *A* value and *S* value of FO membranes become less important due to the low water flux. However, the *A*/*B* ratio shall be kept as high as possible to eliminate reverse solute leakage to feed solutions;(3)The mechanical strength of the FO membrane should not be compromised, although a porous substrate with a small S value is preferred for FO feed with low TDS. Testing *A* and *B* values under low-pressure RO mode is a good method to ensure the general handleability of the membrane;(4)The porous substrate of the FO membrane does not help make a tight polyamide layer. To increase the solute rejection or *A*/*B* ratio of a TFC membrane, creating an interlayer on the porous substrate for interfacial polymerization reaction is an effective approach;(5)Most independent findings suggest that the incorporation of nanoparticles into polyamide layers can increase *A* and possibly A/B when the selectivity-permeability trade-off is properly controlled. However, if the TFC and TFN membranes in the same period are compared together, statistically, the permeability coefficient (*A*) and selectivity (*A*/*B*) of TFN are lower than those of the TFC membrane, which is different from our expectation. Future research may need to take advantage of the excellent water permeability and selectivity of nanomaterials so that TFN membranes truly lead to TFC membranes from a lateral statistical point of view;(6)More pilot and industrial scale studies are needed to guide the development of membranes and module designs. Membrane researchers and developers shall pay more attention to FO applications, which shall be promising and show energy savings compared to other processes/technologies.

## Figures and Tables

**Figure 1 membranes-12-00955-f001:**
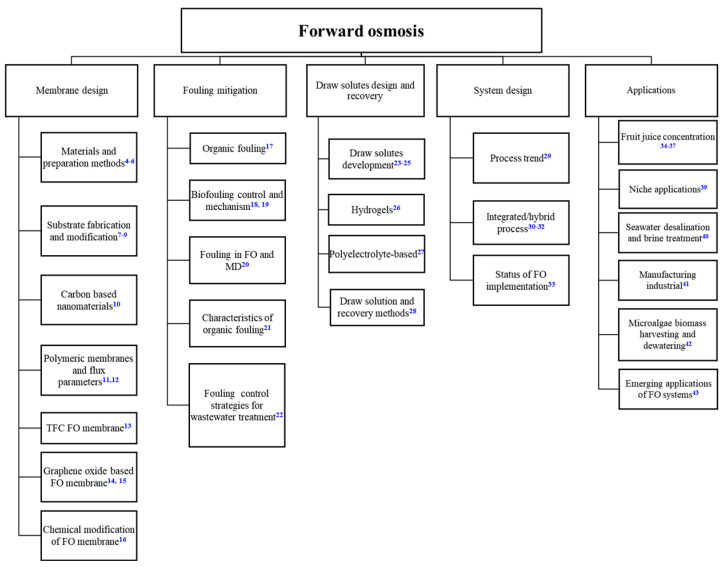
The focus of FO review papers published from 2018–2022. The number in the upper-right indicates the corresponding reference.

**Figure 2 membranes-12-00955-f002:**
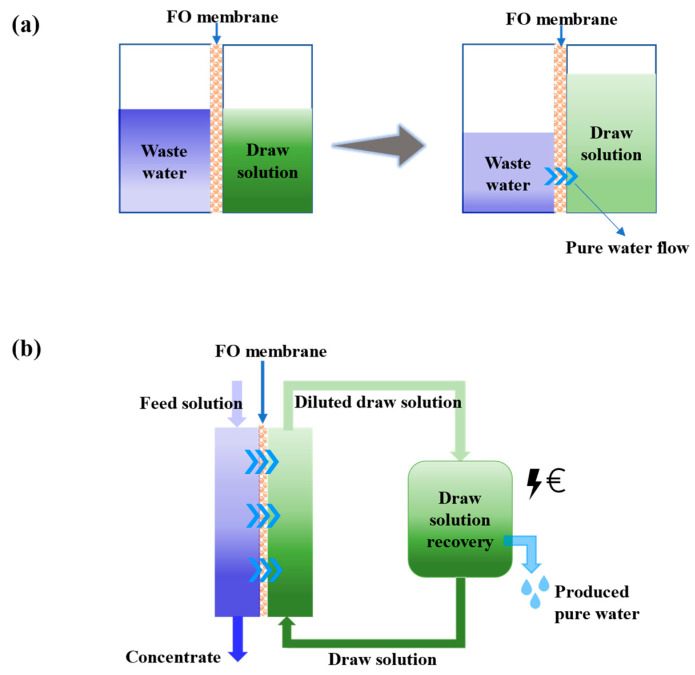
(**a**) Concept of FO and (**b**) a practical FO system coupled with a draw solutes regeneration unit (adapted from [44]).

**Figure 3 membranes-12-00955-f003:**
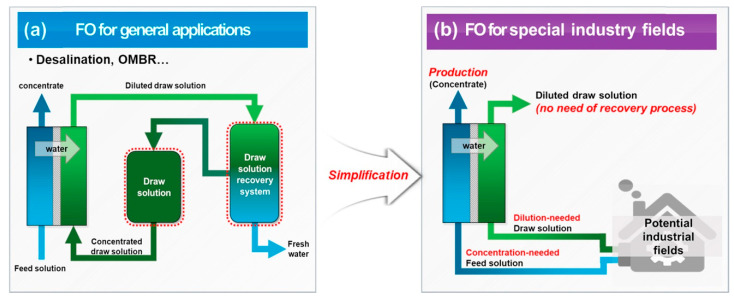
(**a**) FO application for general application with a need for draw solutes recovery; (**b**) FO for special industry fields without the need for a recovery process [56].

**Figure 4 membranes-12-00955-f004:**
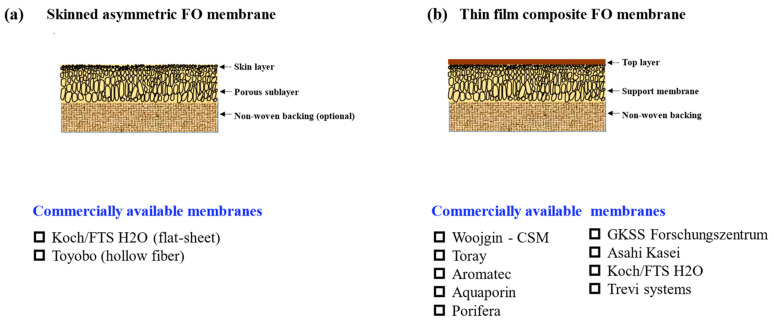
Illustration of the structure disparity of skinned asymmetric membrane (**a**) and TFC membrane (**b**) and the current commercially available membranes of each type.

**Figure 5 membranes-12-00955-f005:**
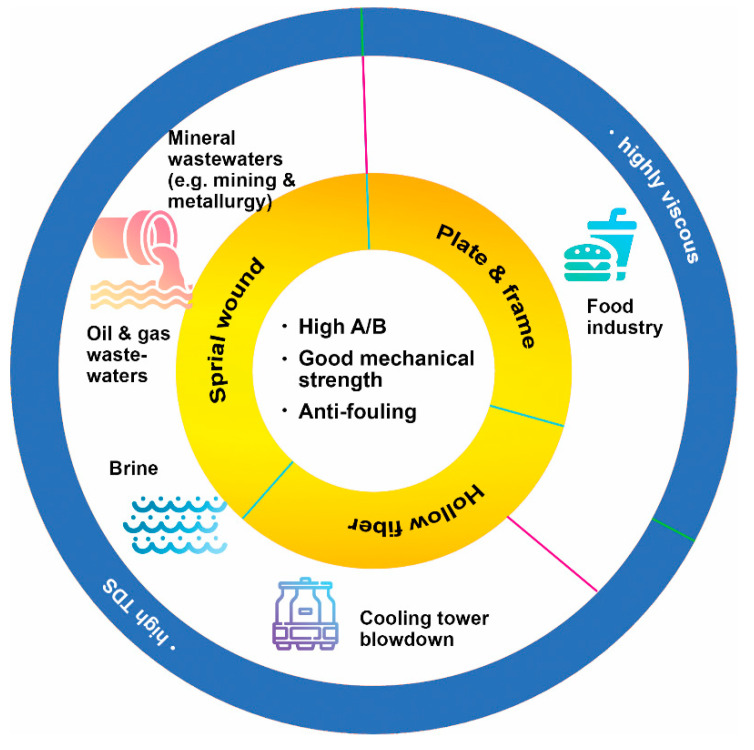
Selection of suitable FO membrane modules for different FO applications.

**Figure 6 membranes-12-00955-f006:**
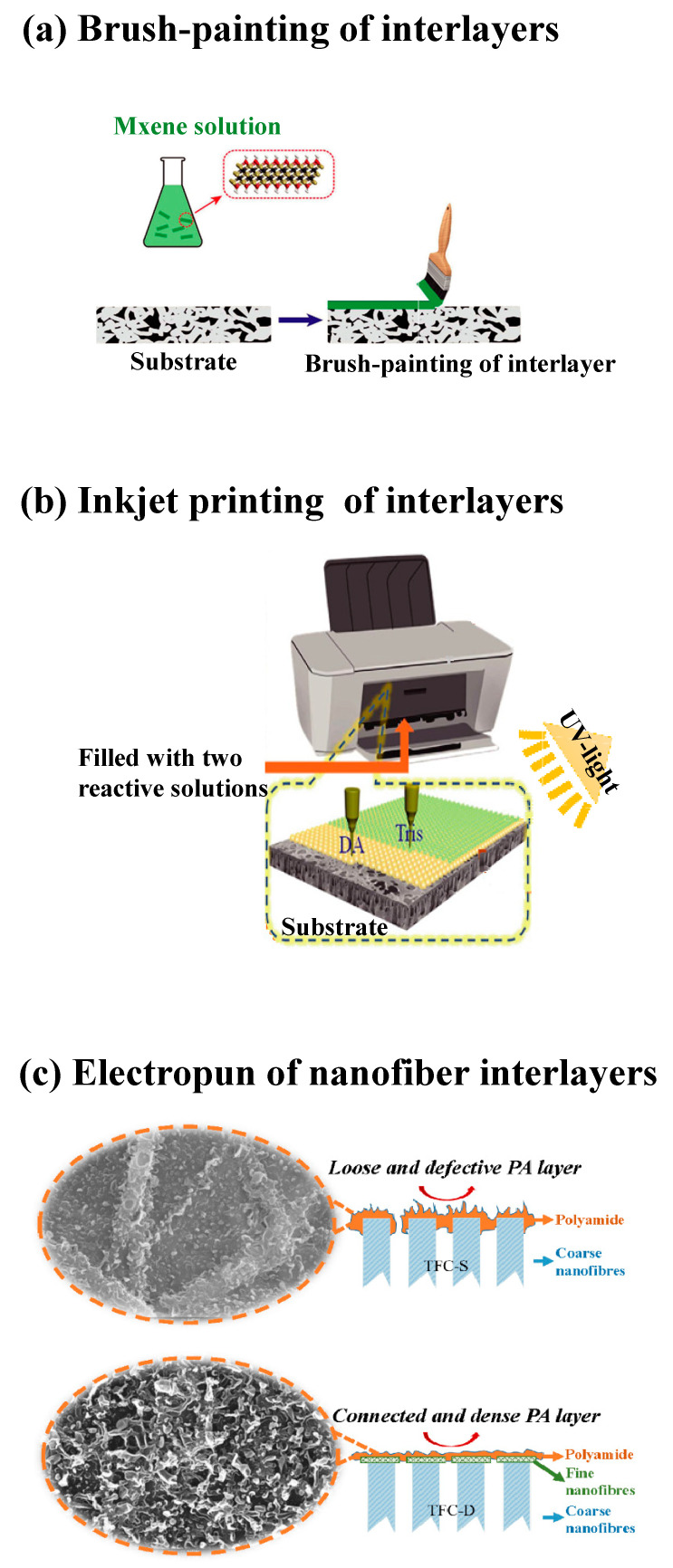
Technology translation from lab to large scale. (**a**) Brush-painting of Mxene on nylon membrane; (**b**) inkjet printing of dopamine and tris solution on UF membrane for mass production; (**c**) Construction of ultrafine nanofiber interlayer via electrospinning. All figures were taken from the references.

**Figure 7 membranes-12-00955-f007:**
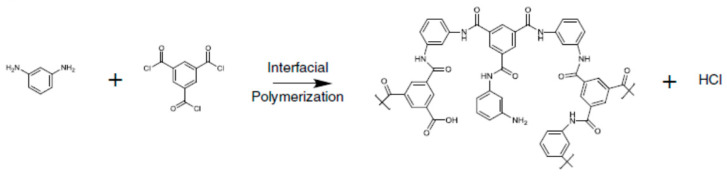
Interfacial polymerization between MPD (aqueous phase) and TMC (oil phase).

**Figure 8 membranes-12-00955-f008:**
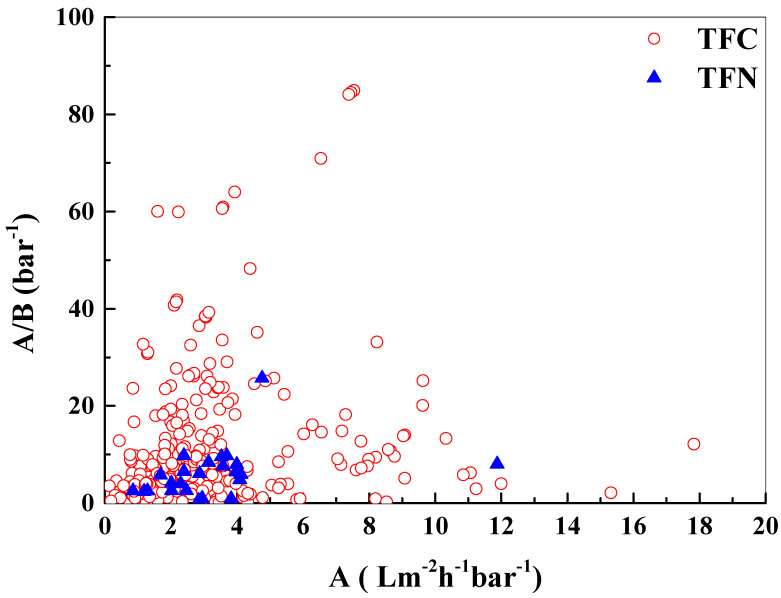
*A*/*B* vs. *A* of the TFC and TFN membranes collected from OMD (The Open Membrane Database).

**Figure 9 membranes-12-00955-f009:**
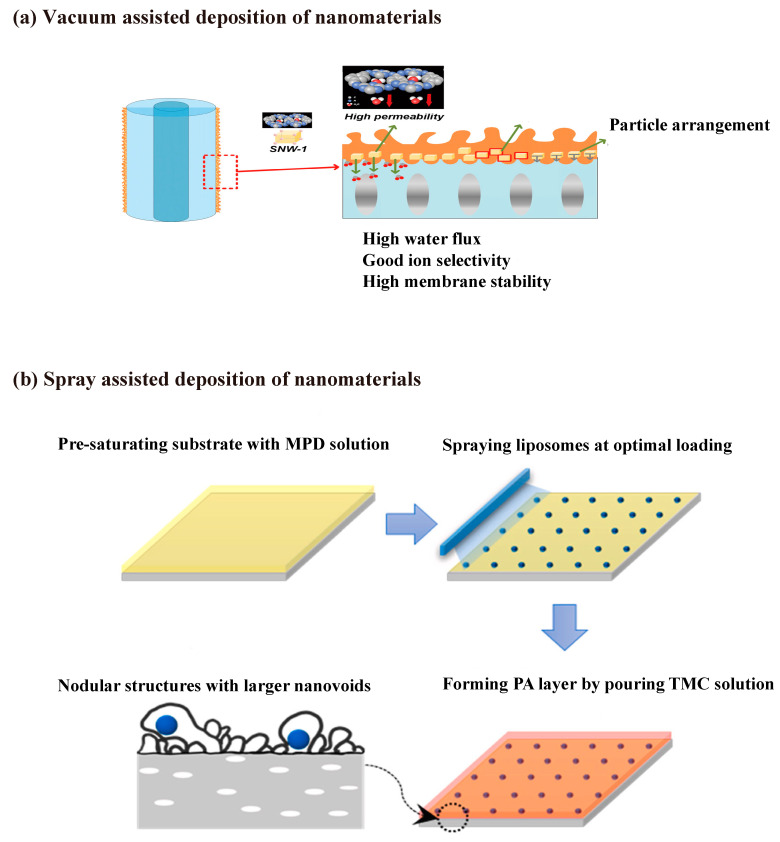
Methods for incorporating nanomaterials in the selective layer. Vacuum-assisted deposition of nanoparticles (**a**) and spray-assisted disposition of liposomes (**b**) into polyamide layer via interfacial polymerization approach. All the figures were taken from the references.

**Figure 10 membranes-12-00955-f010:**
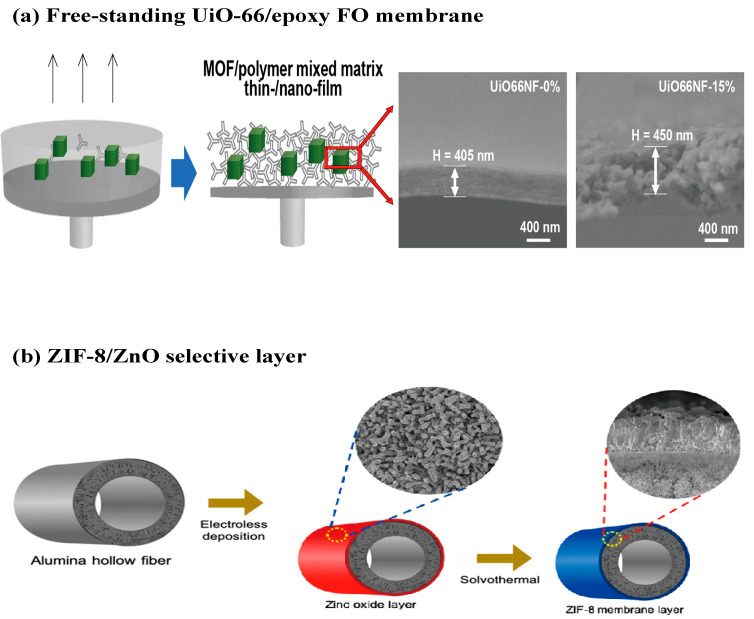
Review of the novel non-polyamide selective layer. (**a**) Freestanding of UiO-66/epoxy FO membrane; (**b**) ZIF-8/ZnO selective layer. All figures were taken from the references [155,156].

**Table 1 membranes-12-00955-t001:** Review of the representative draw solutes in the FO process.

Classification	Draw Solutes	Regenerate Method	Molecular Weight/Size	Js/Jv	Ref.
**Non-responsive:** inorganic salts; polyelectrolytes; surfactants; zwitterions **Features:** widely available; high solubility; difficult to regenerate	NaCl	-	58.5 g/mol	0.76	[59]
	MgSO_4_	-	120.37 g/mol	0.15 g/L	[60]
	MgCl_2_	-	95.21 g/mol	0.22 g/L	[60]
	NMe_4_-Cr-OA	MD	-	-	[61]
	Polydiallyldimethylammonium chloride (PolyDADMAC)	NF	5919 g/mol	0.014 g/L	[62]
	Zn-Bet-Tf_2_N	Solvent extraction	-	-	[63]
	EDTA-2Na	MD	-	-	[64]
	1,4-bis(3-propane- sulphonate sodium)-piperazinediethanesulfonic acid disodium-sulfate	Acid precipitation	105 nm	-	[65]
	Bifunctional zwitterion of (1-(3-aminopropyl) imidazole) propanesulfonate (APIS)	Acid precipitation +filtration	-	-	[66]
**Responsive:** magnetic nanoparticles; volatile liquids; NH_3_-CO_2_; responsive small molecules and polymers **Features**: specifically designed; easy regenerate; cut energy cost	Trimethylamine-carbon dioxide (TMA-CO_2_)	Thermal separation	59.11 g/mol	-	[67]
	Tetrabutylphosphonium p-toluenesulfonate ([P4444]TsO)	∼98% was precipitated by heating the draw solutes at 60 °C	-	0.002815 mol/L	[58]
	Oligo-deep eutectic solvent	Phase separation at 5 °C	-	0.043 g/L	[68]
	Poly(propylene glycol-ran-ethylene glycol) monobutyl ethers (PAGBs)	Thermos responsive lower critical solution temperature (LCST) 42 and 53 °C	1810–3911 g/mol	-	[69]
	Organic-coated engineered superparamagnetic iron oxide nanoparticles	-	12.3 ± 1.0 nm	-	[70]
	Gelatin-coated magnetite nanoparticles (MNPs)	Magnetic field	40 nm	-	[71]
	Nitrogen Rich CO_2_-Responsive Polymers	-	12,000 g/mol	-	[72]
	Pluronic^®^ L35	95 °C	1900 g/mol		[57]

**Table 2 membranes-12-00955-t002:** Basic information of the typical reported FO membranes and testing conditions.

Membrane Key Information	*A*, Lm^−2^ h^−1^·bar^−1^	*B*, Lm^−2^ h^−1^	*A*/*B*, bar^−1^	*R*, %	*P*, Bar	*S*, µm	J_v_, Lm^−2^ h^−1^ (AL-FW)	J_s_/J_v_, g/L	Draw Solution (Feed Water) ^(^#^)^	Surface Velocity, cm/s	[Ref]
HTI-CTA	0.59 ± 0.04	0.36 ± 0.05	1.6	88.8 ± 2.1	5	417 ± 41	13.6	0.74	1 M (DI)	16.7	[59]
HTI-TFC	1.48 ± 0.06	0.35 ± 0.01	4.2	94.7 ± 1.5	5	453 ± 52	17.7	0.55	1 M (DI)	16.7	
DPE-TFC	6.7 ± 0.15	0.68 ± 0.02	9.85	98.1 ± 0.2	5	168 ± 4	53.0	0.28	1 M (DI)	16.7	
NIPS-TFC	1.86 ± 0.2	0.77 ± 0.14	2.4	91.6	5	796 ± 85.3	17.0 ± 0.9	-	2 M (DI)	-	[84]
NIPS-TFC	4.00 ± 0.33	0.22 ± 0.05	18.2	96.7	2	290 ± 56	26	0.14	1 M (DI)	10–40	[85]
NIPS-TFC	1.21 ± 0.01	0.12 ± 0.02	10.1	93.6 ± 2.4	2	240.5	22.1	0.19	1 M (DI)	-	[86]
NIPS-TFC	2.12	5.35	0.4	91.4	5	484	21.3	0.23 *	1 M (DI)	-	[87]
Nanofiber TFC	2.99 ± 0.11	0.41 ± 0.12	7.3	74.2 ± 3.9	0.51 *	174	42	<0.25	1 M (DI)	~9	[78]
Nanofiber TFC	2.82 ± 0.10	0.50 ± 0.02	5.6	77.2	7	187.9	40.64	-	1 M (DI)	11.27	[88]
TFC	0.58 ± 0.01	0.05	11.6	91.1	10	200 *	12.3	-	0.5 M (DI)	2.5	[89]
CNTs hollow fiber TFC	2.45 ± 0.10	0.12 ± 0.04	20.4	92.6 ± 1.4	1	125.6	61.0	0.14	1 M (DI)	25	[90]
TFN	4.47 ± 0.24	0.81 ± 0.01	5.5	96.7 ± 0.2	2	741	11.4	0.27 ± 0.04	1 M (DI)	4.9	[79]
TFN	5.1 ± 0.13	0.39 ± 0.03	13.1	90.9 ± 0.7	2	-	30.2	0.35	1 M (DI)		[91]
TFN	2.5	-		92.5	2	58.6	39.2	0.1	1 M (DI)	8	[81]
TFN	2.55 ± 0.01	0.19 ± 0.02	13.4	96.8 ± 0.4	3	-	12.9	0.11	0.5 M MgCl_2_ (DI)	7.3	[82]

(1) # Draw solute is NaCl if not specified; (2) * denotes the calculated or transformed data; (3) % if not specified means the weight percentage. (4) NIPS is an abbreviation for non-solvent-induced phase separation.

## Data Availability

The data in Figure 8 was obtained from https://openmembranedatabase.org/reverse-osmosis-database, on 5 September 2022.

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
