# Peer review of "Forward Osmosis Membranes: The Significant Roles of Selective Layer"

_membranes, 2022, doi:10.3390/membranes12100955_

Round 1

Reviewer 1 Report

The manuscript "Forward osmosis membranes: the significant roles of selective layer"has an interesting and actual subject of the research field. The presented results are very interesting and useful.

However, the presentation is not focused on the promising title, having a slightly more general character.

Thus, either adjust the title or expand accordingly the part of the manuscript related to the selective layer!

The references regarding the selective layer are few compared to the rest of the bibliography!

The diagram from figure 1 would be very interesting to be correlated with bibliographic references in each rectangle.

Generally, the figures have a good design, but sometimes they are harder to read (ex. figure 6)!

Author Response

Thanks for reviewer. Please find our responses to your questions and the revised manuscript as attached.

Reviewer 2 Report

Tian et al. presented an insightful review on the role of a selective layer in forward osmosis applications. The focus was on achieving high selectivity. The porous substrates used in high-performance FO membrane fabrication were also discussed. It was also concerned that the nanomaterials in the selective layer affected the membrane performance. Overall, from the publication point of view, the manuscript is well-written, covering essential aspects of the topic. Having said that, I have the following comments:

1.      English grammar and expression need to be examined by the native English speaker.

2.      The section on nanomaterials application in the selective layer should be elaborated in more detail. Critical discussion is expected in this section. Emerging nanomaterials should be discussed, and how fabrication conditions can be optimized to achieve even better performance.

3.      The summary should be written more elaborately.

4.      Outlook should include a viewpoint on nanomaterials selection and optimizing process thereof. 

Reviewer 3 Report

The article is generally good and can be accepted after the following corrections:

1- Scientific articles are written in the Third-Person format. Thus, remove all pronouns, such as we, he, ...

2- There are many grammar and spelling mistakes 

3- Summary and outlook should be changed into Conclusion
